# Differences in Self and Proxy Assessments of Self-Determination in Young People with Intellectual Disability: The Role of Personal and Contextual Variables

**DOI:** 10.3390/bs13020156

**Published:** 2023-02-11

**Authors:** Cristina Mumbardó-Adam, Clara Andrés-Gárriz, Alberto Sánchez-Pedroche, Giulia Balboni

**Affiliations:** 1Department of Cognition, Development and Educational Psychology, Faculty of Psychology, University of Barcelona, 08035 Barcelona, Spain; 2Faculty of Psychology, Education and Sport Sciences, University Ramon Llull, 08022 Barcelona, Spain; 3Department of Applied Pedagogy and Educational Psychology, Faculty of Education, University of Balearic Islands, 07800 Ibiza, Spain; 4Department of Philosophy, Social Sciences and Education, University of Perugia, 06123 Perugia, Italy

**Keywords:** self-determination, intellectual disability, adolescents, informants, teachers

## Abstract

Background: Assessing self-determination in students with intellectual disabilities (IDs) is a primary step in facilitating progress monitoring. Researchers have developed both self and proxy assessments to favor a more in-depth evaluation of self-determination expression. However, to date, limited research has explored the congruence between both assessments. Methods: To address this need, the present study analyzes the differences between 219 adolescents with ID; 63% being males with an age range from 13 to 21 years (M = 16.8; SD = 1.72); and their teachers in their assessment of self-determination and explores which factors (students’ age, sex, level of ID and opportunities at school) might explain those differences. The participants were recruited intentionally. Students with IDs completed two questionnaires: the AIR Self-Determination Scale and the Spanish version of the Self-Determination Inventory, which was also completed by their teachers. Results: Significant differences were found in the self-determination assessment, with teachers rating it lower. Further, students’ sex and the opportunities they were provided at school to engage in self-determined actions were found to explain the differences in self-determination assessment. Conclusions: Research and practice initiatives to assess self-determination in young people with IDs must consider that informants’ points of view might be influenced by students’ sex and by contextual opportunities to engage in self-determined actions. Implications for further research and practice are discussed.

## 1. Introduction

Self-determination has largely been stated as a crucial construct in everyone’s life [1] and has been related to positive outcomes for people with and without disabilities [2]. Further, enhancing the skills associated with self-determination (i.e., decision-making, goal-setting, and problem-solving) has been shown to improve postsecondary education [3] and employment and independent living outcomes [4]. Recent research has also shown that promoting individuals with disabilities’ self-determination increases their quality of life [5,6]. This construct has been defined as a “dispositional characteristic manifested as acting as the causal agent in one’s life” [1] (p. 258) under the Causal Agency Theory approach under which this study is based [1].

Self-determination is actually composed of three essential characteristics and seven component constructs: volitional action, agentic action, and action–control beliefs [1]. Acting with volition implies that the person sets their own goals based on personal preferences and initiates action when they decide to do so. Acting agentically requires regulating and managing actions toward goal achievement and navigating obstacles and problems as they occur. Further, these actions are mediated by action–control beliefs that embody self-beliefs about one’s own abilities and how empowered one feels to act on and reach their goals. Further, these essential characteristics that define self-determined actions are operationalized through the acquisition of skills associated with self-determination, such as decision-making, problem-solving, goal-setting, and goal attainment, among others. All of those are examples of skills related to self-determination that come into play in everyone’s life and are crucial to reaching personal goals and acting as a causal agent of one’s life, including the lives of people with intellectual disabilities (IDs).

Research has tried to pinpoint if personal factors, such as age, sex, or ID severity, influence the expression of self-determination-related skills. Concerning age, self-determination seems to develop across developmental stages, with levels of self-determination increasing throughout adolescence [7]. Regarding sex, to date, the findings remain inconclusive, with studies highlighting women with IDs as showing a higher degree of self-determination [8], but other studies finding no significant differences [9]. Regarding ID severity (based on participants’ Intelligence Quotients), students with moderate levels of ID usually obtain significantly lower scores on self-determination than their peers with mild ID [9]. In parallel, people with IDs and a higher level of support needs have also reported lower levels of self-determination [10].

As a personal and dispositional trait, self-determination-related skills are put into action across contexts that shape the person’s tendency to act in a self-determined manner, thereby enabling the individual to navigate challenges and opportunities as they arise. When the context of young people with IDs, such as teachers or families, provides instruction and opportunities for developing these skills, these youth engage in self-determined action, thus contributing to the development of self-determination [11].

Research in this field has blossomed in recent decades, with investigators trying to delve into how self-determination impacts people with IDs and identify mechanisms to propel its promotion. As a primary step for self-determination improvement, practitioners and stakeholders must be able to measure this construct and monitor its baseline, change and further development. For instance, measuring self-determination is crucial to understanding the support that needs to be provided to scaffold its progress. As a result, researchers have built instruments to assess self-determination across countries, such as the Arc Self-determination Scale [12] or the AIR Self-determination scale [13], this latter being used in this study to measure opportunities to engage in self-determined actions at school. Both instruments have been largely used and validated in different languages and countries. Both are self-report measures. An appropriate assessment of people with ID self-determination should use a variety of methods, putting the person with ID at the center of the process. However, a third-party perspective, that is, the information their relatives and other stakeholders as support services staff, might also provide useful information [14,15]. For instance, recent studies have highlighted the need to use both self and proxy measures of quality of life in individuals with IDs during the transition to adulthood [16].

This type of proxy reports measure might be the only one available for people with high levels of support needs and with difficulties in language expression and comprehension that might not be otherwise assessed. Further, when combined with self-reports, proxy reports provide essential information on those informants’ perspectives. The knowledge those informants have on the self-determination construct, as well as the importance given to self-determination, can determine the degree to which the skills related to self-determination are promoted [17]. Teachers tend to evaluate their students’ capacity for self-determination lower than the students themselves [18,19]. In fact, when teachers rate self-determination instruction as important, they provide more opportunities for its development [20].

Opportunities to act in a self-determined manner have consistently been stated as a relevant factor in self-determination development [10,21,22]. For example, young people with disabilities have been found to have more opportunities to engage in self-determined actions at school compared to their peers without disabilities [23]. Self-determination enhancement is inextricably linked to the opportunities youth with IDs are provided to learn and practice self-determination-related abilities. In addition, seeing students engage in self-determined actions nurtures third-party perceptions of students’ self-determination capacities [24]. In turn, when teachers believe in their students’ capacities, they provide more opportunities to engage in self-determined actions. Consequently, as previously stated, self-determination assessment is clearly affected by respondents’ perspectives.

Given the role that third-party perspectives can play in self-determination development, researchers have recently endeavored to develop self-determination proxy reports. Stemming from the Causal Agency Theory in which this study is based, two instruments have been developed to date: the Self-Determination Inventory [25] and the AUTODDIS scale [14,26]. The Self-determination Inventory (SDI) [23] is a self-report that targets young people from 13 to 22 years old with and without disabilities and is composed of a student and a parent or educator version. Both versions of this instrument have been used to evaluate self-determined actions in this study. The AUTODDIS scale [14,26] assesses the self-determination of young people and adults with IDs (from 11 to 40 years old) from a third-party perspective and has been developed within the Spanish context. A recent study conducted in the U.S. focused on exploring the relationship between students’ and teachers’ scores on the SDI, with the findings suggesting low correlations between self- and proxy scores [19]. Teacher respondents tended to report that adolescents had lower levels of self-determination, although further research is needed in additional contexts to confirm this trend.

The present study thus intends to provide further evidence of self-determination assessment differences amongst youth with IDs and their teachers and disentangle the variables that might contribute to these differences. ID severity has consistently been related to self-determination levels, with people with higher levels of support needs scoring lower in terms of self-determination, although this relationship might be influenced by other variables, such as having opportunities to engage in self-determined actions [22].

We hypothesize that teachers’ self-determination ratings will be lower than self-reports, which is consistent with previous research [19], and that both the ID severity of students with IDs and self-reported opportunities to engage in self-determined actions might predict the differences between teachers’ and students’ assessment of self-determination. The present study thus intends to respond to the following research questions.

(RQ1) Are there differences in the self and proxy assessments (third-party point of view) of self-determination in young students with ID?

(RQ2) Which personal (ID severity, age, and sex) and contextual factors (opportunities at school) predict differences in self-determination assessment?

## 2. Materials and Methods

### 2.1. Participants and Setting

The participants were adolescents with IDs and their teachers. Although the data were collected with more participants (298), some were deleted to meet the prerequisites of data analysis (see below). The participants were thus 219 adolescents with IDs (63% males) and were aged 13 to 21 years (M = 16.8; SD = 1.72). They were recruited in 23 special education schools across Spain; that is the context where most students with IDs are enrolled after primary school. Participants’ primary diagnoses were mild (29%), moderate (48%), and severe (23%) IDs. Severity was identified either with the assessment of intellectual functioning or with both adaptive behavior and intellectual functioning assessment, as some of the assessments were made before adaptive behavior was considered a key component of ID diagnosis. As all of the students underwent an official assessment upon enrollment, ID and comorbid disabilities were diagnosed by clinical professionals, psychologists, or psychiatrists, who used existing protocols for diagnosis. All of the participants had comorbid disabilities, the most prevalent ones being learning disabilities (34%), behavioral disorders (15%), and attention deficit and hyperactivity disorder (11%). Other disabilities were autism spectrum disorder (9%), physical disabilities (4%), language and communication disorders (4%), and visual (1%) and hearing (2%) impairments.

The teachers who responded to the proxy assessment on self-determination were those who taught students that participated in this study. Sixty-four teachers provided information for 1 to 12 of their students. The headmasters of the schools that received the participation request contacted the teachers and students’ families (as explained below) and decided not to provide demographic information about their teachers (such as sex and age).

### 2.2. Instruments

#### 2.2.1. The Self-Determination Inventory Student Version (SDI:SR)

Two versions of the SDI are available, the student and the educator versions [25] and both were used in this study. The original and Spanish versions [23] of the SDI are rooted in the Causal Agency Theory [1] and are intended to measure the essential characteristics and associated component constructs of self-determined actions. The U.S. student pilot version, upon which this translation is based, has 45 items and is divided into three essential characteristics and seven component constructs [27]. The volitional actions essential characteristic has 13 items and gathers information related to autonomy (6 items) and self-initiation (7 items). The agentic actions (10 items) essential characteristic includes self-direction (6 items) and pathways thinking (4 items) and refers to the ability to self-regulate and monitor progress while working toward goals. Finally, action–control beliefs (22 items) include control expectancy (9 items), psychological empowerment (7 items), and self-realization (6 items) and encompass one’s self-knowledge of their capacities and the abilities that are used to reach a goal. To answer each item, the participants moved a cursor on a slider bar that marked their position between ‘‘I disagree’’ and ‘‘I agree’’. The more the student moved their cursor to the right, the more he/she agreed with the corresponding statement. The slider bar captured numbers from 0 to 100. Previous research has established three second-order (volitional action, agentic action, and action control beliefs) and seven first-order factorial structures with construct validity analysis, demonstrating good fits of the theoretical model (χ^2^/df = 3.045, CFI = 0.942, TLI = 0.953, SRMR = 0.106) [23]. Cronbach’s alpha for the data collected with this scale in this study was 0.886, and McDonald’s Omega was 0.893, confirming a high internal consistency.

#### 2.2.2. The Self-Determination Inventory Parent or Educator Version

The parent or educator versions mirror the student version’s structure (7 subdimensions and 3 higher dimensions, as explained above). It is composed of 38 items written all in the third person, except for 7 items of the action–control beliefs component. The items that have been deleted from the educator version tend to express students’ perceptions, such as “it is better to be yourself than to be popular”, and thus are not suitable from a third-party perspective assessment. In this instrument, the respondents must give their third-party point of view instead of the estimation of the individual with IDs point of view. This version has not yet been validated in the Spanish context, but evidence of its validity and reliability for this study has been gathered. In terms of reliability, Cronbach’s alpha for this scale was 0.927, and McDonald’s Omega was 0.934, stressing a high internal consistency. In terms of validity, evidence of the factor structure of this instrument was collected in this study and presented below.

To gather evidence of validity, first following expert suggestions [28], the independent and dependent variables were analyzed to detect the presence of (a) univariate outliers or cases with values higher than z > |3.29| and (b) multivariate outliers or cases with extreme values based on the Mahalanobis Distance. The normality of the (c) univariate and (d) multivariate distribution of continuous variables was analyzed by, respectively, computing the asymmetry and kurtosis indexes and Mardia’s test. While no univariate outliers were found, multivariate outliers (70) were identified and excluded from the subsequent analyses. Each time a multivariate outlier was detected and excluded, the self-determination total score distribution was normalized. This procedure was repeated until no outliers were detected. Therefore, the subsequent analyses were conducted on the answers provided by the remaining 219 participants.

A confirmatory factor analysis (CFA) and an explorative structural equation model (ESEM) were run to verify if the factor structure of the educator version of the SDI followed the same factorial structure as that of the student version. The 7-factor structure of the self-determination instrument under the Causal Agency Theory was assessed first by CFA and then by the ESEM approach [29]. ESEM provides additional information as the factor loadings of both the observable items and the latent variables can be reported, whereas, in the CFA measurement models, the factor loadings are fixed at zero. ESEM uses a less restrictive measurement model and allows the items to load freely in the seven self-determination dimensions. The estimation of the goodness of the model was based on the following fit indexes: root-mean-square error of approximation (RMSEA), standardized root-mean-square residual (SRMR), comparative fit index (CFI), and Tucker–Lewis index (TLI) [30]. Values greater than 0.95 for CFI and TLI and smaller than 0.05 for RMSEA and SRMR suggest a reasonable fit [31].

The CFA analyses to assess the SDI educator version structure resulted in indexes that were quite adequate according to the criteria of goodness of fit previously described: RMSEA (CI) = 0.046 (0.039–0.052), SRMR = 0.056, CFI = 0.882, TLI = 0.872. Conversely, the ESEM 7-factor solution showed better and fairly acceptable goodness for all the values fit indexes: RMSEA (CI) = 0.037 (0.028–0.046), SRMR = 0.051, CFI = 0.948, TLI = 0.920. Additionally, according to the factor loading ≥ |0.30|, 35 of the 38 items were loaded in at least one of the seven factors. In addition, except for three items, all of the remaining items were loaded in their theoretical factor. Differently from the original structure, the second item of pathways thinking (“The student finds other ways to get things done.”) loaded in the self-direction factor. The first (“The student is confident in his/her abilities.”) and second items (“This student knows his/her strengths.”) of self-realization were loaded into the empowerment factor. We should note that these three items remained in the theoretical second-order factor, as pathways thinking and self-direction are part of the agentic actions’ domain, and self-realization and empowerment belong to the action–control beliefs domain.

#### 2.2.3. The AIR Self-Determination Scale (AIR-S)

The AIR-S measures a person’s capacities and opportunities for self-determination and is available in student, educator, and parent versions [13]. For the purposes of this study, the Spanish online version of the student form (AIR-S) was used to measure the students’ capacity and opportunities for self-determination [32]. The AIR-S has 24 questions divided into two scales that gather data relating to students’ self-reported capacities and opportunities to engage in self-determined actions. The capacity scale is further divided into two subscales and covers questions about students’ (1) ability related to self-determination (Ability subscale) and (2) perceptions about performing self-determined actions (Perception subscale). The opportunity scale is also composed of two subscales that measure (1) students’ perceptions of their opportunities to perform self-determined actions at home (Opportunities at Home subscale) and (2) at school (Opportunities at School subscale). The scores are rated on a Likert scale from 1 (Never) to 5 (Always). The AIR has been extensively used in the field and has been shown to have adequate test–retest reliability (0.74 after 3 months) and a strong internal consistency (split-half test = 0.95; Cronbach’s alpha ranging from 0.89 to 0.99). In terms of validity, the original authors conducted a factor analysis that supported a four-factor structure explaining 74% of the instrument variance [13]. This instrument has been validated in the Spanish context for youths with and without disabilities, showing good model fit indices (CFI = 0.982, TLI = 0.962, RMSEA = 0.043), and invariance measurement amongst youth with and without disabilities also held [32]. For the purpose of this study, only the opportunities scale was used.

### 2.3. Procedure

The Ethics Committee of the University Ramon Llull approved the present study. The research team intentionally contacted headmasters of special education schools by email across six different autonomous communities from Spain (Catalonia, the Community of Madrid, Aragon, Balearic Islands, the Valencian Community and Castile and Leon). Thus, the sampling method was intentional. Once headmasters affirmatively responded to the invitation email, the first author met with the teachers that the headmasters had previously considered for participation in the study either by phone or through video call. In these meetings, the first author trained the teachers on how to support their students to answer the instruments, if needed, and also explained how the proxy reports should be fulfilled. As answers to the instruments were provided online, to be included in this study, the schools were required to provide computers for teachers and students to complete the assessment and an Internet connection for the online administration of the instrument. Twenty-one special education schools agreed to participate in the study and met the requirements mentioned above. A sample of the questionnaires was sent so the teachers could intentionally choose students with disabilities aged 13 to 22 years who could provide reliable information (i.e., sufficient language/reading comprehension to understand the items with support as necessary). To be included in this study, students’ parents or guardians had to provide consent for participation (if under the age of consent), and teachers and students themselves had to provide assent to participate. Additionally, the inclusion criteria for teachers providing information about their students’ self-determination included having provided direct services to their students for at least 6 months.

Consistent with the scale administration protocols, the teachers explained item meanings and the structure of the response system to support the administration. The students could be provided with needed support per administration guidelines, including facilitating access to information (i.e., reading the questions) and understanding questions (i.e., giving synonyms of challenging words). Teachers had to answer the questionnaire by scales by providing their third-party perspective on their students’ self-determination. In 9 cases, the proxy respondents were missing (more than 15% of the items were missing, and thus we considered there were too many missing answers), and they were retrieved from the analysis.

### 2.4. Data Analysis

As the SDI educator version showed an acceptable measurement structure that mirrored the SDI student version, analyses aimed at responding to the research questions of the study were performed. Standardized data and a significance level of 0.05 were used for this analysis. Data analyses were performed using SPSS (version 27) with a total sample of 219 cases.

First, correlations and intra-class correlations (two-way random model, total agreement type) were used for the total self-determination score and the dimensions measured with the SDI (self and proxy versions). Standardized measures were used in both cases. The means and standard deviations for self and proxy respondents were computed. To compare the means of both groups, and considering that self and proxy versions of the SDI did not have the same number of items, a stable and comparable measure was calculated by dividing each dimension and self-determination total score by the number of items. A *t*-test for matched samples was also used to compare self-determination and self-determination dimensions differences in assessments, both measured with the self and proxy measures of the SDI. Effect sizes (Cohens’*d*) were also computed. Second, the predictive power of youth with ID characteristics (severity of ID, sex, and age) and opportunities provided at school for self-determination, measured with the AIR Self-determination scale, over the difference between informants in terms of the total self-determination score was tested through hierarchical regression analysis.

Before running the regression, the multicollinearity between the predictors was analyzed by surveying the Tolerance index and the Variance Inflation Factor [30]. The normality, linearity, and homoscedasticity of the residuals were explored by analyzing the shape of the residuals’ distribution and comparing the residuals’ scatterplot to the theoretical values provided in [28] for the set of predictors and separately for each predictor. The independence of errors was investigated via the Durbin–Watson test, and the presence of outliers in the solution was detected by analyzing the standardized residuals.

## 3. Results

### 3.1. Comparison between Self and Proxy Reports

Correlations and intra-class correlations between self and proxy scores were statistically significant but poor for the total scale score (ICC = 0.254, *p* < 0.001) and for the rest of the dimensions (Table 1). Significant statistical differences were also found amongst the total self-determination scores (*t*_(218)_ = 9.139, *p* < 0.001) with a large effect size (*d* = 0.618) and the self-determination dimensions, with small (*d* = 0.318) to large (*d* = 0.732) effect sizes depending on the respondents. The students and teachers thus reported very different scores of student self-determination, with students always scoring higher than teachers.

### 3.2. Personal and Contextual Variables on Self and Proxy Assessment Differences

The assumptions of the regression analyses of linearity, normally distributed residuals, homoscedasticity, and the absence of multicollinearity were met for this model. The linear regression analysis showed that personal characteristics of youth with ID and opportunities that these youth report to engage in self-determined actions within the school context predicted the differences between informants in terms of the total self-determination scores. Table 2 shows the results of the hierarchical regression analysis with age, sex, ID severity (first step), and opportunities to engage in self-determined actions at school (second step) as independent variables, and the differences between teacher and self-report self-determination measure as the dependent variables. First, the combination of the three personal characteristics variables explained 7.2% of the variance of the differences in the assessment of self-determination (*R*^2^ = 0.072; *F*_(3,218)_ = 5.531, *p* < 0.001), although just sex (*β* = 0.252; *p* < 0.001) was found to have a significant impact in predicting those differences. Interestingly, the teachers scored female students higher than males in terms of overall self-determination, whereas male students scored themselves higher than their female peers in terms of overall self-determination scores. However, when the opportunities at school measured with the AIR self-determination scale was entered, the variance explained increased to 11.1% (*R*^2^ = 0.111; *F*_(3,218)_ = 6.649, *p* < 0.001), and both sex (*β* = 0.263; *p* < 0.001) and opportunities at school (*β* = 0.199; *p* = 0.003) predicted self-determination assessment differences. Neither ID severity nor age had a significant impact on the self-determination assessment differences. The squared semi-partial correlation coefficient (*sr*^2^) suggested that both variables contributed to self-determination assessment prediction.

## 4. Discussion

The main aim of the presented study was to delve into the differences in the self and proxy assessments (third-party point of view) of self-determination in young students with IDs. In addition to understanding the differences in self and proxy assessments, this study is, to the best of our knowledge, the first that tries to comprehend which factors influence those differences. Significant differences were found in students’ and teachers’ assessments of self-determination, thus implying that both respondents assessed self-determination differently. Students with IDs consistently scored higher than teachers when they self-assessed their self-determination, thus implying that students with IDs considered they have a higher level of self-determination than their teachers’ assessment, consistent with previous research [19]. When trying to understand which factors explained this low correlation, students’ sex and opportunities to engage in self-determined actions were decisive. No other personal factor (age or ID severity) contributed to explaining the assessment differences. Interestingly, teachers scored female students higher than males in overall self-determination, whereas male students scored themselves higher than their female peers in overall self-determination scores. Further, students’ perceptions of having more opportunities to engage in self-determined actions contributed to explaining those differences as well. Concretely, higher opportunities to express self-determination were associated with higher differences in teachers’ and students’ ratings.

The limited agreement between adolescent and teacher ratings of self-determination that has been stated in students with IDs [19], but also in students with emotional and behavioral disorders and learning disabilities [18], stresses the need for ongoing research to interpret these differences. Further research would then have to tackle the differences in the self-determination conceptions among students, teachers, and other educational agents to better align future intervention proposals and initiatives. Additionally, that those differences were nuanced by students’ sex is a finding that merits further attention. The role of sex in self-determination expression has traditionally been contradictory [33]. Researchers have found conflicting impacts of sex, with some researchers finding females with IDs showing higher levels of self-determination [8], whereas other results have been inconsistent [9]. Interestingly though, in the present study, when teachers assessed their students’ self-determination, they considered females to have higher self-determination than males [8]. However, when students evaluated their own self-determination, males considered themselves to have higher levels than their female peers, thus highlighting that females tend to assess their own capacities lower than others consider them to be, consistent with previous research [34]. Females are more prone to characterize themselves in more stereotypic terms, for example, as less agentic and less competent in leadership [34], for instance, as less self-determined than their male peers. As such, educators must focus on empowering women to reduce the gender stereotypes (allocated to their students’ sex), guiding their behaviors and nurturing their beliefs, as biased conceptions definitely impact their self-determination.

Maladjusted conceptions about capacities and skills hinder a person’s use of these skills. Concretely, female students’ conceptions of having fewer self-determination capacities might reflect on their own engagement in self-determined actions. Conversely, male students’ conceptions of having more capacities related to self-determination might end up disrupting what they want and actually can do, thus heightening the frustration of these students. Future research should then, in the first place, further study the role of sex in self-determination expression and, perhaps, more importantly, consider sex in the intervention initiatives, given the impact this might have in the future expression of self-determined capacities. Consistently, implications for practice derived from these results highlight teachers’ responsibilities to foster a more adjusted and tailored self-knowledge of their students’ capacities when supporting their self-determination development. As previously argued, nurturing an adjusted and realistic sense of empowerment and self-knowledge is crucial for students’ action–control beliefs, one of the essential self-determination characteristics [11].

Thus, teachers play an essential role in promoting their students’ self-knowledge and self-determination skills. In addition, the opportunities students think they have at school also explain the differences between students’ and teachers’ assessment of their self-determination. Research has consistently proved that contextual opportunities favor people with IDs self-determination [10,21,22]. Acting in a self-determined manner necessitates a favorable context that supports this self-determination expression instead of hindering it. When students think and see that their contexts favor their self-determination, they feel empowered to act in a self-determined manner. As such, seeing their students engaging in self-determined actions raises teachers’ beliefs about their students’ capacities [15]. For this reason, further research and practice initiatives must investigate how to build contextualized opportunities mediated by teachers for students’ self-determination expression.

This study is not exempt from some limitations that must be considered when interpreting the results. First, although the information was gathered on students’ personal information, teachers’ demographic information, such as age or sex, was not provided by the headmasters as to preserve their privacy. Second, other personal and contextual variables that might explain those differences were not collected in this study. We must consider the possible effect of the sociocultural context in relation to the differences in self- and proxy assessments of students with IDs self-determination. The gendered stereotypes mark the roles, behaviors, and expectations attributed to men and women, which have a very strong cultural nuance and may vary drastically depending on the cultural context [35]. Considering the importance of context in the development of self-determination [36], our results must always be contextualized (see, for example, the role of the sociocultural background; [37,38]). Third, there may be other variables not considered in this study that might influence the differences exposed, so the presented variables must not be considered the only factors able to influence the perceptions of self-determination. Additionally, fourth, all of the students with IDs and their teachers that participated in this study were recruited in special education schools and thus might not represent the reality of students enrolled in mainstream schools. However, it must be noted that the vast majority of students with IDs that reach secondary school in Spain are typically enrolled in special education schools.

Despite these limitations, the present study highlights important considerations regarding the assessment of self-determination in adolescents with IDs. It shows the different perceptions in the assessment of self-determination between students and teachers [19], as well as some variables that might explain those differences. For this reason, both practitioners and researchers must approach its assessment from multiple ways, including self and proxy evaluations, but also other strategies, such as observation within the contexts where self-determination develops. Consistently, research and practice initiatives to assess self-determination in young people with IDs must consider that informants’ point of view might be influenced both by the person’s sex and its attributed roles and also by how this person is provided with contextual opportunities to engage in self-determined actions and actually uses them. All in all, more qualitative and in-depth research is needed to better support both the assessment and promotion of self-determination in young people with IDs. Self-determination is definitely a complex construct whose expression is nuanced by personal and contextual variables, and thus context must be given the importance it deserves in evaluation processes.

## Figures and Tables

**Table 1 behavsci-13-00156-t001:** Correlations, Intra-class Correlations, *t*-Student test between Matched Samples and their Effect Sizes between Self-determination and Self-determination Dimensions Proxy and Self-responses.

	Self *M(SD)*	Proxy *M(SD)*	*r*	ICC	*t-Student*	*Cohen’s d*
Self-determination	6.91(1.56)	5.59(1.9)	0.254 **	0.254 **	9.139_(218)_ **	0.618
Autonomy	7.07(1.81)	6.33(2.00)	0.249 **	0.250 **	4.701_(218)_ **	0.318
Self-initiation	6.47(2.03)	5.53(1.99)	0.195 *	0.195 *	5.414_(218)_ **	0.366
Self-direction	6.83(2.00)	5.31(2.20)	0.249 **	0.250 **	8.741_(218)_ **	0.591
Pathways thinking	6.54(2.16)	4.80(2.38)	0.182 *	0.183 *	8.877_(218)_ **	0.600
Control Expectancy	6.95(1.88)	5.65(1.96)	0.178 *	0.179 *	7.800_(218)_ **	0.668
Empowerment	7.11(1.81)	5.81(2.18)	0.193 *	0.198 *	7.565_(218)_ **	0.370
Self-knowledge	7.27(1.89)	5.31(2.42)	0.251 **	0.252 **	10.839_(218)_ **	0.732

Note: ** *p* < 0.001, * *p* < 0.01.

**Table 2 behavsci-13-00156-t002:** Hierarchical Multiple Regression of Personal Characteristics, Opportunities at School and Self-Determination Assessment Differences for Youth with Intellectual Disability.

	*R* ^2^	*ΔR* ^2^	*ΔF*	*β*	*sr* ^2^
**Model 1**	0.072	0.059	5.531 **		
ID severity				0.078	0.078
Sex				0.252 **	0.251
Age				−0.045	−0.044
**Model 2**	0.111	0.094	6.649 **		
ID severity				0.063	0.063
Sex				0.263 **	0.261
Age				−0.059	−0.058
Opportunities at school				0.199 *	0.197

Note: ** *p* < 0.001, * *p* < 0.01.

## Data Availability

Data are available upon request to the corresponding author.

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
