# Peer review of "Differences in Self and Proxy Assessments of Self-Determination in Young People with Intellectual Disability: The Role of Personal and Contextual Variables"

_behavsci, 2023, doi:10.3390/bs13020156_

Round 1
Reviewer 1 Report
Overall a well written article. It had good coherence and structure. A few tips are provided to improve the article.
- In the abstract section of the article, the background is long and in the method section, please add the population, sample and sampling method.
- In the method section of the article, please write the society and the sampling method accurately and clearly.
-Most of the article references are old. Please bring most of the references of the last 5 years
Reviewer 2 Report
This is a strong paper that addresses an important topic, highlighting the difference in self-reporting and teacher reporting assessments of adolescents with ID. More must be learned about these differences and how to rectify these differences. The conclusion regarding the findings about the differences in females self rating compared to males, and teacher ratings is also significant and is particularly compelling.
I support the motivation for this study, and feel the findings are very compelling. I do work with this population and feel the findings accurately reflect my experience. I was not aware of the male /female differences in self-reporting and find that result very compelling and hope the researchers continue to research this aspect of their findings for both the potential ethical and patriarchal issues it might uncover. I would appreciate a little more insight about this finding be added to this manuscript.
Reviewer 3 Report
I think the paper is very interesting. The results are congruent with the objectives of the study.
The discussions should be more expanded and I would like to better understand the differences of this research with the existing literature. The references should also be increased.
Reviewer 4 Report
The manuscript entitled "Differences in self-assessment and proxy assessment of self-determination in young people with intellectual disability: the role of personal and contextual variables" is an interesting and relevant article about not only the differences between the own perspectives of young people with ID and the perspective that their teachers have on the self-determination of the former, but also analyzes the role of relevant personal and contextual variables, which have been frequently pointed out in the literature in relation to the self-determination. It is also an interesting article due to the size and characteristics of the sample, as well as the validity and reliability of the instruments used. In general terms, I consider that it is an article that represents an important contribution to our field of study and that it has a solid theoretical and methodological foundation. For this reason, I would like to congratulate the authors and recommend that it be accepted for publication in Behavioral Sciences with some minor modifications. Next, I will expose the aspects that could be improved or clarified.
[Abstract]
1. Please, provide some specific data about the size of the sample: N people with ID (gender; age mean and SD). It is not totally clear in the abstract and along the manuscript if the same young people who completed the self-report were evaluated by their teachers or if both groups differ totally or partially.
2. It was surprising for me, after reading the title, to discover that only one contextual variable was analyzed. That’s the reason why I think that the title is very attractive but maybe not very descriptive. The authors might consider that maybe listing the specific variables (age, gender, level of ID and opportunities) in the title might be more adequate and make this manuscript easier to be found by others. However, I understand that following this suggestion the title would be too long. In this case, I suggest the authors to clearly state in the abstract the personal and contextual variables that were analyzed).
[Introduction]
3. When the authors talk about personal factors, they point out “sex”, but I would say that most of the research do not talk about sex but about gender. Maybe a clarification about this is needed (if they talk about sex as synonymous of gender, but I think that would not be correct). In the same way, when talking about level of ID, it would be recommendable that the authors specify what they mean with level of ID (previous research is based on IQ, on adaptive behavior, or both? Is it based on the level of support needs? The explanation about the influence of level of support needs is briefly stated at the end of the introduction, but I think it should be included together with the scientific literature review of the rest of personal variables.
4. Why two measures of self-determination (ARC and AIR) were used? This issue might be better clarified in the introduction.
[Participants]
5. I think the information or explanation about ID diagnosis is redundant (second paragraph says something that was already explained in the first one).
6. Please, clarify if participants and teachers were asked about sex or about gender. Specify clearly if the students evaluated by teachers where the same students who completed the self-report.
[Instruments]
7. Given that the validation of the instruments is not a goal of this manuscript, I think that the evidences about the validity (CFA) should be provided within the “Instruments” section (as reliability was described here), not in the Results section.
[Procedure]
8. I am not sure if I understand what the authors mean when they say that “in 9 cases, the proxy respondents were missing (more than 15% of the items were missing) and they were retrieved from the analysis.”. ¿In 9 cases there were too many missing answers? ¿Or there were no proxy-reports for 9 students who did complete the self-report?
[Results]
9. The subsection titled “Principal results” would benefit of having a more descriptive title, such as “Relation between self and proxy reports” or something similar. I would add another subsection to describe the results related to the second question or hypothesis (i.e., influence of personal and contextual variables).
[Discussion]
10. All students were selected in special education schools. I think this is an important limitation that should be acknowledged in this section.
[Data Availability Statement]
11. There is a misprint errata in “autho” (the “r” is missing)
[References]
12. The references are very pertinent and well selected, although there seemed to be a certain inbreeding in them. Given that the authors have published a lot about this topic, self-citations are very understandable and pertinent, but the manuscript could benefit from citing works by other groups, especially when they refer to other related constructs, such as quality of life.
13. It would be convenient to take advantage of this extension of citations with the inclusion of references to articles published especially in the last two years.
14. References to articles published in Spanish should include the translation of the title into English (for instance, those published in Siglo Cero).
